

# An evaluation of the importance of spatial resolution in a global climate and hydrological model based on the Rhine and Mississippi basin

Imme Benedict[1], Chiel C. van Heerwaarden[1], Albrecht H. Weerts[2,3], and Wilco Hazeleger[1,4]

[1]Meteorology and Air Quality Group, Wageningen University, Droevendaalsesteeg 4, 6708 BP Wageningen, The Netherlands
[2]Hydrology and Quantitative Water Management Group, Wageningen University, Droevendaalsesteeg 4, 6708 BP Wageningen, The Netherlands
[3]Deltares, P.O. Box 177, 2600 MH Delft, The Netherlands
[4]Netherlands eScience Center (NLeSC), Science Park 140, 1098 XG Amsterdam, The Netherlands

*Correspondence to:* I. Benedict (imme.benedict@wur.nl)

**Abstract.**

The hydrological cycle of river basins can be simulated by combining global climate models (GCMs) and global hydrological models (GHMs). The spatial resolution of these models is restricted by computational resources and therefore limits the processes and level of detail that can be resolved. To further improve simulations of precipitation and river-runoff on a global scale, we assess and compare the benefits of an increased resolution for a GCM and a GHM. We focus on the Rhine and Mississippi basin. Increasing the resolution of a GCM (1.125° to 0.25°) results in more realistic large-scale circulation patterns over the Rhine and an improved precipitation budget. These improvements with increased resolution are not found for the Mississippi basin, most likely because precipitation is strongly dependent on the representation of still unresolved convective processes. Increasing the resolution of vegetation and orography in the high resolution GHM (from 0.5° to 0.05°) shows no significant differences in discharge for both basins, because the hydrological processes depend highly on other parameter values that are not readily available at high resolution. Therefore, increasing the resolution of the GCM provides the most straightforward route to better results. This approach works best for basins driven by large-scale precipitation, such as the Rhine basin. For basins driven by convective processes, such as the Mississippi basin, improvements are expected with even higher resolution convection permitting models.

## 1 Introduction

Hydrometeorological extremes present a combination of atmospheric and hydrological processes. On a global scale, these processes are simulated by coupling global climate models (GCMs) and global hydrological models (GHMs). With that, we can forecast and give future projections of the hydrological cycle and its extremes. However, the spatial resolution of climate and hydrological models limits the detail level that can be resolved in a numerical simulation. With higher spatial resolution, and therefore an improved representation of physical processes and level of detail, we expect more accurate results when modelling the impact of climate on hydrological processes. However, restricted computer capabilities limit these improvements. Currently,



the common horizontal resolution of GCMs is around 150 km (CMIP5; Taylor et al. 2012). For GHMs this resolution is around 50 km (ISI-MIP; Haddeland et al. 2011; Schellekens et al. 2016; Beck et al. 2016b).

To improve the detail level at catchment scale, it is a dilemma whether to invest computer power in high resolution global models or in regional downscaling. High resolution global climate models lead to better resolved large-scale processes (Scaife
et al., 2011; Jung et al., 2012; Demory et al., 2014; Hodges et al., 2011), cyclones (Strachan et al., 2013; Manganello et al., 2012) and more pronounced small-scale extremes. More benefits of solving the climate at global high resolution can be found in Haarsma et al. (2016) and the references in that article. For hydrological modelling, an increase in resolution leads to improved spatial heterogeneities in topography, soil, and vegetation (Wood et al., 2011) and therefore results in more realistic surface runoff and evaporation. However, increasing the resolution of a GHM also requires to increase the number of unknown, and
often not easily quantifiable, model parameters. This brings in large uncertainties when modelling hydrology across multiple spatial scales. There are multiple ongoing initiatives that assess the benefits of global models with very high spatial resolution for both the atmosphere (HighResMIP CMIP6; Meehl et al. 2014; Haarsma et al. 2016), and in hydrology (Wood et al., 2011; Bierkens et al., 2015).

In parallel to the research on global modelling, hydrological studies often use downscaled products to study regional climate
variations and their hydrological impact (Jacob et al., 2014), as the spatial resolution of a basin can be substantially increased compared to a global model. Although dynamical downscaling has many benefits, it is not able to reduce biases that are related to errors in large scale circulation patterns (Van Haren et al., 2015), which are related to the low-resolution GCMs used as boundaries conditions for the downscaled products (Hazeleger et al., 2015; Fowler et al., 2007; Wood et al., 2004).

Here, we compare low and high resolution runs of a global climate model, as well as a global hydrological model, to
systematically study the effect of an increased spatial resolution. By comparing all combinations of resolutions (Table 1), we aim to answer our main research question: what are the benefits of an increased resolution and are they similar for a global climate and global hydrological model?

We analyse three main components of the hydrological cycle: precipitation, evaporation and discharge. Thereby, we focus on the Rhine and Mississippi basin, where long measurement records are available for validation. We have chosen these two
basins as their climatic drivers are different, which can contribute to our understanding of the processes resolved with increased spatial resolution. The precipitation budget of the moderately sized Rhine is determined by the mid-latitude storm track, which are shown to be better represented with higher resolution models (e.g. Van Haren et al., 2015; Zappa et al., 2013). On the other hand, the budget of the Mississippi is influenced by moisture input from multiple drivers; moisture is advected from the Pacific, from the Caribean and the Gulf of Mexico and extreme precipitation occurs within tropical cyclones (Fig. 2). In
addition, convective preciptiation plays an important role over the Mississippi basin. The representation of these processes is resolution dependent.

The paper is structured as follows: more details about the basins is given in the next Sect. 2. In Sect. 3 the models, data and methods are described. In the result Sect. we discuss the sensitivity of precipitation, evaporation and discharge to the higher resolution GCM and GHM. Thereafter, we present two hydrometeorological extreme events in some more detail (Sect. 4.4). In
Sect. 5 a discussion on the methodology is given and Sect. 6 gives the summary and conclusion of this study.




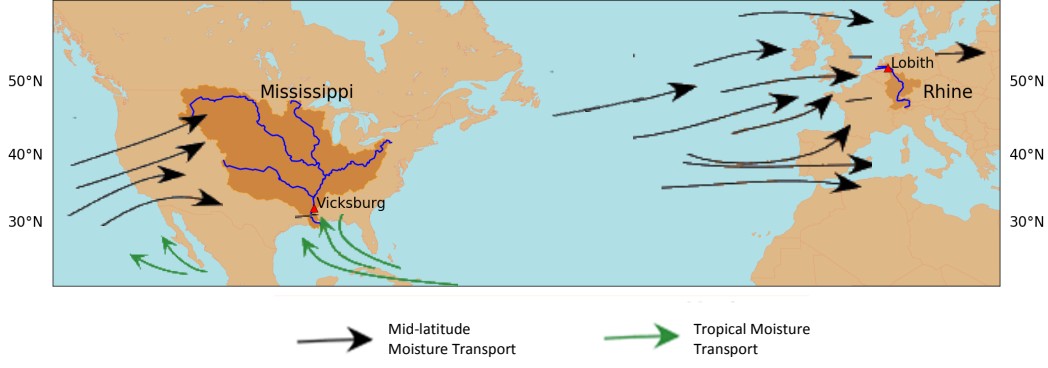

**Figure 1.** Two spatial resolution simulations of the GCM are used to force two different spatial resolutions of the GHM.

**Figure 2.** Map indicating the Rhine and Mississippi basin, the rivers, the used gauge station (Lobith and Vicksburg) and the conceptual location of mid-latitude moisture transport (black arrows) and tropical moisture transport (green arrows). Figure adapted from: http://www.physicalgeography.net/fundamentals/7s.html

## 2   Basin description: Rhine and Mississippi

The river Rhine originates in the Swiss Alps and flows through Switzerland, Germany and the Netherlands where it empties into the North Sea. In this study, we analyse discharge at Lobith, which is the location where the Rhine enters the Netherlands. Therefore the basin is defined upstream of Lobith (measuring about 165 000 km$^2$, see Table 1). The average discharge at
5   Lobith is 2 200 m$^3$ s$^{-1}$ and the highest discharges occur in late winter and spring. During this period large-scale rainfall events associated to storms occur over saturated soils, which can lead to extreme flood events. Snow melt, in combination with frozen soils, can occasionally lead to extreme flood events as well (Hegnauer et al., 2014).

The Mississippi basin is more than sixteen times larger than the Rhine basin. It measures 2 981 000 km$^2$ (Table 1), which makes it the fourth-largest river basin in the world. The Mississippi River originates at Lake Itasca, Minnesota from where it
10   flows South towards the Gulf of Mexico. The two largest tributaries of the Mississippi are the Missouri and Ohio river. Here, we study the discharge of the Mississippi basin at Vicksburg, where the average discharge is 16 792 m$^3$ s$^{-1}$ (Fig. 2). Most flood events occur in winter and spring due to heavy precipitation, snowmelt and rain-on-snow events.



**Table 1.** Basin characteristics of the two study basins including basin area, used gauge station and its average discharge their.

| Basin | Basin area (km$^2$) | Gauge station | Average discharge (m$^3$ s$^{-1}$) |
|---|---|---|---|
| Rhine | 165 000 | Lobith | 2 200 |
| Mississippi | 2 981 100 | Vicksburg | 16 000 |

## 3 Data and methodology

### 3.1 Global climate model EC-Earth

We use the high resolution experiments (Haarsma et al., 2013) from the state-of-the-art atmospheric global climate model EC-Earth V2.3 (Hazeleger et al., 2010, 2012). EC-Earth is based on the European Centre for Medium-Range Weather Forecasts numerical weather prediction model (IFS cy31r1). An improved hydrology scheme (H-TESSEL; Balsamo et al. 2009; Van den Hurk et al. 2000) is inserted in EC-Earth, compared to IFS. Actual evaporation is generated from this scheme by solving the energy balance for specific land tiles. EC-Earth is forced with prescribed sea surface temperatures (SST), based on observations in current climate (NASA data at 0.25° resolution; for details we refer to Haarsma et al. 2013). Observed greenhouse gases and aerosol concentrations are also used as boundary conditions.

The high resolution experiments have a horizontal spectral resolution of T799, which corresponds to 25 km, and 91 vertical levels (further referred to as high and T799). For comparison in resolution, the same model runs are performed with a spectral horizontal resolution of T159, corresponding to 120 km and 62 vertical levels (further referred to as low and T159). The parameterization packages of the high and medium resolution runs are the same (Van Haren et al., 2015). The land-use products are similar for the two resolutions, where the high-resolution information is used for the high-resolution runs. For both resolutions, six members of five years (2002-2006) are created, resulting in 30 years of data representing present climate. It should be noted that the fixed boundary conditions (SST and greenhouse forcing) decrease the independency of the members. More information on the experiment and the spin-up can be found in Haarsma et al. (2013).

### 3.2 Global hydrological model W3RA

W3RA is the global hydrological model that we use in this study. It is based on the landscape hydrology component model of the AWRA system (AWRA-L; Van Dijk 2010a, b). AWRA-L can be considered a hybrid between a simplified grid-based land surface model and a non-spatial, or so-called lumped, catchment model applied to individual grid cells. The model consists of two hydrological response units (HRU's); deep-rooted tall vegetation (forest) and shallow-rooted short vegetation (herbaceous), each of which occupying a fraction of a grid cell. Vertical processes are described for each HRU individually. The model consists of three soil layers and runs with a daily time step. Actual evaporation is calculated with the energy balance. The parameters in W3RA at 0.5° resolution are determined with a regionalization approach (Van Dijk, 2010c). For full technical details about the model algorithm and parameters, we refer to the technical documentation (Van Dijk, 2010b).



We run the model at the original horizontal resolution of 0.5° (∼ 50 km) and at a higher horizontal resolution of 0.05° (∼ 5 km). With this increased resolution, orography and vegetation will be better represented as these parameters are known at high resolution. However, many other parameters are not physically based or difficult to determine at multiple spatial scales. To allow a fair comparison between the two model resolutions, we remapped the parameters from the 0.5° to the 0.05° resolution.

This method is verified by Melsen et al. (2016), who concludes that parameters can to a large extent be transferred across the spatial resolution. Maps of orography and vegetation (division of HRU per grid cell) are used at the 0.05° resolution. The model algorithm is not adapted for the higher resolution, which means that there are no extra processes resolved at the higher resolution.

The resolution of the GHM does not perfectly coincides with the resolution of the GCM (see Table 1). Therefore, we remap
the climate variables in between. Runoff is translated towards discharge using the wflow routing scheme (Schellekens, 2016), which is based on the kinematic wave approximation. For the 0.5° resolution GHM routing is performed at 0.5°. For the 0.05° resolution GHM, we aggregate the runoff data towards a 0.083° resolution to perform the routing on the river network (ldd) maps available from the PCRGLOB-WB model (Van Beek and Bierkens, 2009). For each member, we perform a spin-up of five years (length of timeserie per member). We leave out the first year for analysis due to the influence of spin-up, which
results in 24 years of discharge simulations to analyse.

### 3.3 Verification datasets

We use the E-OBS dataset version 4 (Haylock et al., 2008) from 1985 until 2015 (30 years) for precipitation comparison over the Rhine basin. For the Mississippi basin, the Climate Prediciton Center (CPC) 0.25° Daily US Unified Gauge-Based precipitation dataset (Higgins et al., 2000) is used from 1985-2015 (30 years).

For the verification of actual evaporation, we use the GLEAM (Global Land Evaporation: the Amsterdam Methodology) dataset version 3.0a (Martens et al., 2016) from 1985 until 2015 (30 years). This product is primarily driven by potential evaporation estimates using Priestley-Taylor (Priestley and Taylor, 1972) and by passive microwave remote sensing data.

Daily discharge data for the Rhine at Lobith and the Mississippi at Vicksburg are obtained from the Global Runoff Data Center (GRDC, 2007) from 1985 until 2015 (30 years).

In addition to the observational datasets, we verify our model results with reanalysis data from the ECMWF. A global atmospheric reanalysis, ERA-Interim (Dee et al. 2011; further referred to as ERAI), is used from 1985 up to 2014 (30 years). ERA-Interim has a spatial resolution of around 80 km and 60 vertical levels (T255L60) and is based on IFS release Cy31r2 (comparable to Cy31r1 used in the EC-Earth simulations), which includes the land-surface TESSEL scheme (Viterbo and Beljaars, 1995). In addition, the ERA-Interim LAND reanalysis (Balsamo et al., 2013) is shortly addressed, where precipitation
from ERA-Interim is corrected with satellite data and an improved land-surface scheme H-TESSEL is used (Balsamo et al., 2009). ERA-Interim LAND is only available until 2010 and therefore we analyse the timeseries from 1985 until 2010. Lastly, the ERA20C dataset (Poli et al., 2016) is used for extra verification of the precipitation budget over the Mississippi (1985-2010). ERA20C is a different reanalysis product also based on IFS (cy38r1) which performs the assimilation on fewer variables than ERA-Interim.





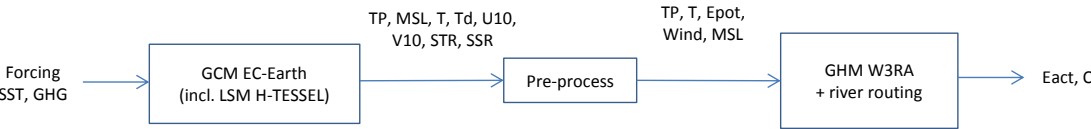

**Figure 3.** Flowchart illustrating the methodology of this study including the global climate model, the global hydrological model and the related variables: total precipitation (TP), mean sea level pressure (MSL), temperature at 2 meter (T), dewpoint temperature at 2 meter (Td), wind component x-direction at 10 meter (U10), wind component y-direction at 10 meter (V10), surface solar radiation (SSR), surface thermal radiation (STR), potential evaporation (Epot), actual evaporation (Eact) and discharge (Q).

### 3.4 Experimental set-up

We use the low and high resolution GCM EC-Earth to force the low and high resolution GHM W3RA (Table 1). To verify our model results, we also force the GHM with ERAI data. The coupling of the GCM and the GHM is illustrated in Fig. 3. We use the following variables from the GCM: total precipitation (TP), mean sea level pressure (MSL), temperature and dewpoint temperature at 2m (T and Td), wind at 10m (U10 and V10), and surface solar and thermal radiation (SSR and STR). In the pre-process phase, potential evaporation (Epot) is calculated using Penman-Monteith (FAO method; Monteith et al. 1965). Then we use potential evaporation, precipitation, temperature, mean sea level pressure and wind to force the GHM.

In this study, we analyse the three main components of the hydrological cycle: precipitation, evaporation and discharge. First, we analyse precipitation from the GCM, because it is the main and most uncertain forcing variable for hydrological applications (Biemans et al., 2009; Fekete et al., 2004). To get a first impression, we compare simulated and observed spatial distributions of 30-year average daily precipitation sums over the basins. Figure 2 indicates the basin areas. With the monthly averages of basin averaged precipitation, we compare the seasonal cycle of the observations with the two resolution GCMs and the ERA-Interim data. The robustness of these results are indicated with 95 % confidence intervals which are obtained after bootstrapping the daily data (Efron and Tibshirani, 1994), assuming all years to be independent.

Additionally, we perform an extra analysis over the Mississippi basin to better understand the precipitation patterns. We focus on the Mississippi, as extensive analysis has already been performed for the Rhine (Van Haren et al., 2015). We analyse the large-scale circulation patterns over the basin and we quantify the convective part of precipitation, which plays an important role in the Mississippi.

Furthermore, we statistically assess precipitation extremes by calculating the return time of annual maxima 10-day precipitation sums (Haren et al., 2013; Shabalova et al., 2003; Kew et al., 2011). We have chosen to analyse 10-day precipitation sums, as multi-day precipitation extremes are mostly connected with extreme discharge (Disse and Engel, 2001; Ulbrich and Fink, 1995). The maxima are rank-ordered and an empirical distribution is applied to determine their return time $T$: $T = m/(N+1)$, where $m$ is the rank-ordered maxima and $N$ is the number of years in the data (30 years). Gumbel plots show the seasonal 10-day precipitation maxima as a function of the Gumbel variate $x = -\ln(-\ln(T))$, which can be translated into a return time





**Table 2.** Four different discharge measures: $\overline{Q}_{\text{mean}_h}, \overline{Q}_{\text{max}_h}, \overline{Q}_{\text{min}_h}$ are respectively the mean, maximum and minimum daily discharge of year number h, ranging from 1 to 24. The total number of years (H) is 24. $\overline{Q}_{\text{peak}_h}$ is the month in which the discharge peak occurred in year h.

| Measure | Explanation | Calculation |
|---|---|---|
| $\overline{Q}_{\text{mean}}$ | 24-year average mean annual discharge (m$^3$ s$^{-1}$) | $\overline{Q}_{\text{mean}} = \frac{1}{H} \sum\limits_{h=1}^{24} Q_{\text{mean}_h}$ |
| $\overline{Q}_{\text{max}}$ | 24-year average annual maximum discharge (m$^3$ s$^{-1}$) | $\overline{Q}_{\text{max}} = \frac{1}{H} \sum\limits_{h=1}^{24} Q_{\text{max}_h}$ |
| $\overline{Q}_{\text{min}}$ | 24-year average annual minimum discharge (m$^3$ s$^{-1}$) | $\overline{Q}_{\text{min}} = \frac{1}{H} \sum\limits_{h=1}^{24} Q_{\text{min}_h}$ |
| $\overline{Q}_{\text{peak}}$ | 24-year mode of month in which yearly discharge peak occurs | $\overline{Q}_{\text{peak}} = mod(Q_{\text{peak}}) \sum\limits_{h=1}^{24}$ |

$T$ in years. The plots are made for annual maxima in every season (DJF, MAM, JJA and SON). These Gumbel plots are only based on 30 data points, which should be taken into account during the interpretation of these plots.

Second, we analyse actual evaporation which couples the physical climate system and hydrology as it can constitute a feedback between the atmosphere and the land surface. Therefore, actual evaporation is calculated within the global climate and global hydrological model, which allows us to compare the two models. We derive monthly averages of basin-averaged actual evaporation over the basins. We only show Eact results from the 0.5° resolution GHM. For comparison, we also show the actual evaporation results from ERAI, from the hydrological model forced with ERAI, and actual evaporation from an independent reference product GLEAM.

Third, we compare monthly averaged discharge from the GHM with observations at Lobith (Rhine) and Vicksburg (Mississippi). In addition, we compare four discharge measures as defined in Table 2: $\overline{Q}_{\text{mean}}, \overline{Q}_{\text{max}}, \overline{Q}_{\text{min}}, \overline{Q}_{\text{peak}}$. Finally, we determine the return times of annual maxima discharge per season, by using the same Gumbel distribution as described for precipitation. It should be noted that these results are based on 24-years of discharge simulations.

In addition to the analysis of these three variables, we aim to better understand the relation between precipitation and discharge. Therefore, we show scatterplots of daily discharge against previous 10-day precipitation sums for both basins and the high and low resolution GCM simulations. In these scatterplots, we only show the discharge results from the 0.5° GHM. The correlations are calculated for each season (DJF, MAM, JJA and SON) and we also include annual maxima. All above described methods statisically compare the variables for different model products. However, we also want to give an indication of the performance of the high resolution model for individual extreme events. Therefore, we will shortly describe one extreme event for the Rhine and one for the Mississippi, by showing the rainfall-runoff response and the synoptic pattern over the basin.





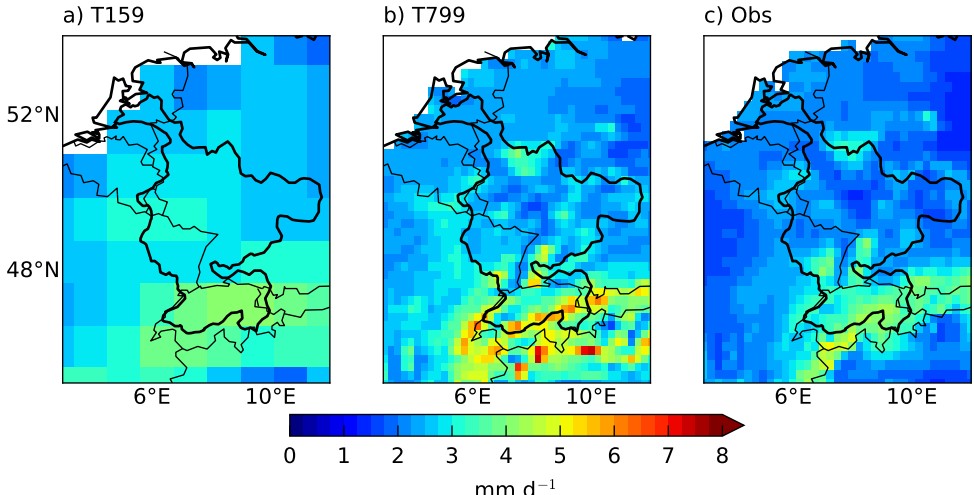

**Figure 4.** 30-year average of daily precipitation sums (mm d$^{-1}$) over the Rhine basin for a) the low resolution EC-Earth simulations (T159),
b) the high resolution EC-Earth simulations (T799) and c) the E-OBS dataset (Obs).

# 4 Results and discussion

## 4.1 Precipitation

### 4.1.1 Precipitation in the Rhine basin

The EC-Earth simulations and the observations (E-OBS) show a similar spatial distribution of precipitation over the Rhine
basin (Fig. 4), with more precipitation over the Alps (4-5 mm d$^{-1}$) than downstream over Western Germany (1-2 mm d$^{-1}$).
The high-resolution model shows, as expected, a more detailed distribution of precipitation over the basin. A higher resolution
orography leads to clearly visible spatial structures such as the Alps, Ardennes and Black Forest within the high resolution
model. At the locations with large precipitation amounts, slight overestimations are found with the high resolution model (Fig.
4b). It is unclear if these overestimations are related to model performance or to underestimation of precipitation in the E-OBS
dataset, which is based on a sparse gauge network in mountainous areas (Hofstra et al., 2009).

Monthly and basin averaged daily precipitation sums of both simulations overestimate the observed precipitation amounts
(Fig. 5). From March until July the high-resolution model outperforms the low-resolution one. Van Haren et al. (2015), who
used the same EC-Earth simulations, found similar improvements in high-resolution precipitation for the region that spans
the Rhine and Meuse basin. They attributed this to the better represented storm tracks over Europe in the high-resolution
simulations and therefore a more accurate horizontal moisture transport. Nevertheless, despite the improvement by resolution,
precipitation is still overestimated from January until June in both simulations compared to the observations and ERAI (Fig.
5a).





**Figure 5.** The upper four panels show monthly averages of basin-averaged daily precipitation sums (mm d$^{-1}$) and daily evaporation sums (mm d$^{-1}$) over the Rhine basin (left panels) and over the Mississippi basin (right panels). The basin and monthly averages of precipitation produced by the convection scheme (CP) are also indicated in dotted lines for the Mississippi. In the lower two panels monthly averages of discharge at Lobith (Rhine, left panel) and Vicksburg (Mississippi, right panel) are shown. The red and blue lines indicate the simulations from respectively the high resolution (T799) and low resolution (T159) GCM. In black we show the observations and the green line shows ERA-Interim data. The dashed lines indicate the simulations with the 0.5° GHM and the dotted lines simulations with the 0.05° GHM. The shaded bands indicate the 95 % confidence intervals computed using bootstrapping.



**Figure 6.** Gumbel plots of seasonal (DJF, MAM, JJA and SON) maximum 10-day precipitation sums [mm] over the Rhine (left panels) and maximum discharge [m³ s⁻¹] at Lobith (right panels) and their related return times T expressed in standardized Gumbel variate $x = -\ln(-\ln(T))$. Observed discharges are shown in black, high resolution forcing (T799) in red, low resolution forcing (T159) in blue and forcing with ERA-Interim in green. The discharge results are output from the 0.5° GHM.





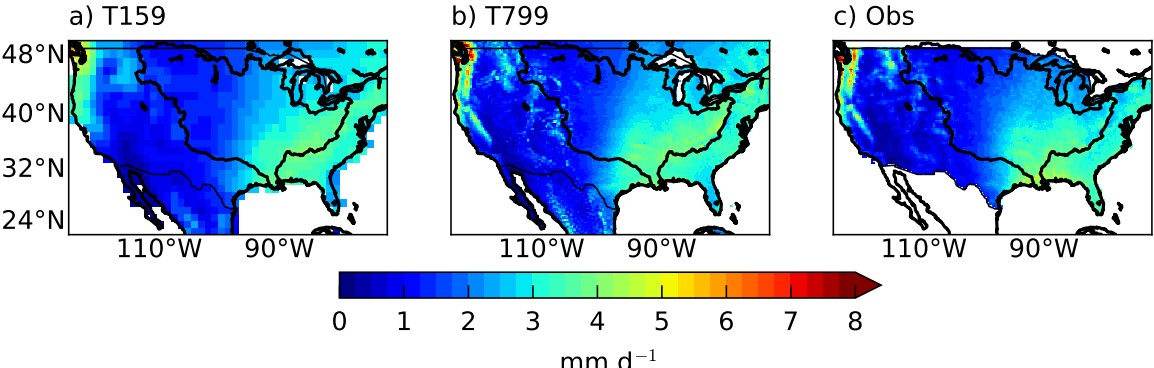

**Figure 7.** 30-year average of daily precipitation sums (mm d$^{-1}$) over the Mississippi basin for the low resolution EC-Earth simulatons (T159), the high resolution EC-Earth simulations (T799) and the CPC dataset (Obs.)

Figure 6 (left panels) shows the influence of model resolution on the return time of annual 10-day precipitation maxima per season. During all seasons, and particular in DJF and MAM, there is a distinct overestimation of precipitation by EC-Earth at lower return times (smaller than two years). This is in agreement with the overestimaton in the monthly averages of precipitation (Fig. 5a). At larger return times (larger than two years), we find an underestimation of precipitation in the GCM data in DJF (Fig. 6a). The extremes in the storm-track season (SON) are quite well reproduced by the model. By comparing the two model resolutions, we find that in MAM and JJA the high resolution model outperforms the low resolution one for all return times, which suggests that with an increased resolution the right large-scale conditions are present to activate convection.

### 4.1.2 Precipitation in the Mississippi basin

Where precipitation over the Rhine is dominated by the storm-track, the Mississippi basin has multiple climatic drivers (Fig. 4). Moisture is advected from the Pacific resulting in high precipitation amounts over the Rocky mountains (4-5 mm d$^{-1}$). The Great Plains, which are situated on the lee side of the Rockies, are very dry (1-2 mm d$^{-1}$), whereas the South-East of the USA is relatively wet because of convection and advection of moisture from the warm tropical Caribbean and Gulf of Mexico. Figure 7 shows the distribution of precipitation over the Mississippi basin for the two resolutions of the GCM and the observations (CPC). There are clear improvements in the distribution of precipitation for the high resolution model, especially over mountain ranges as the Rockies, the Cascades, and the Sierra Nevada, which is in line with previous resolution studies with an atmosphere-only GCM (Duffy et al., 2003) and a coupled ocean-atmosphere GCM (van der Wiel et al., 2016).

Comparison of the simulations with observations reveals an overestimation of precipitation in the North-East of the catchment (Fig. 7). Monthly and basin averaged daily precipitation sums of both simulations show a shift of one to two months in the seasonal cycle, where the highest monthly values occur in April/May instead of in June (Fig. 5b). Moreover, the amount of



precipitation in this shifted peak is overestimated (Fig. 5b). The increase in precipitation in October-November is not observed but occurs, most pronounced, in the high-resolution simulations. A similar peak in October-November is found in the convective part and suggests a bias in convection in the high-resolution model. Similar precipitation biases are found in the EC-Earth simulations for the sub-basin averages (Missouri and Arkansas-Red, not shown). In contrast to the EC-Earth simulations, pre-

cipitation from ERAI shows the correct seasonal cycle (Fig. 5b). EC-Earth and ERAI are based on the same atmospheric model (IFS), albeit different versions. Therefore we hypothesize that the precipitation bias found with EC-Earth, is not present in the ERAI reanalysis, because of the data assimilation process. The precipitation budget from the ERA20c reanalysis data, where assimilation is performed on fewer variables than ERAI, shows a larger bias with observations compared to ERAI (data not shown), supporting our hypothesis.

Apart from the precipitation bias between EC-Earth simulations and observations, no substantial differences in basin-averaged precipitation between the low and high resolution simulations were found (Fig. 5b). This similarity of the two resolutions could be explained by the convective component of precipitation, which is modelled at the sub-grid scale (i.e. parameterized) for both resolutions. We will further discuss convection in the next Sect. (4.1.3). Thereby, we will also assess the sensitivity of resolution to the large-scale circulation over the Mississippi basin.

The bias between observations and simulations is also reflected in the Gumbel plots of 10-day precipitation sums per season over the basin (left panels Fig. 8). In MAM, there is an overestimation of the extremes for all the return times and in JJA an underestimation for all the return times. In SON, there are much larger precipitation extremes in the high resolution compared to the low resolution (Fig. 8). This could possibly be related to the improved simulation of tropical cyclones with higher resolution, although this should be investigated further. In DJF, we find larger biases with the high-resolution compared to

the low-resolution, although previous studies show improvements of extreme precipitation with increased resolution (Iorio et al., 2004; Wehner et al., 2010; van der Wiel et al., 2016; Duffy et al., 2003). In the winter season moisture advection from the Pacific plays a large role. An improved orography could therefore lead to enhanced precipitation amounts. In addition, 'observed' precipitation products, like the CPC dataset, severely underestimate precipitation over the western mountain ranges (Lundquist et al., 2015; Henn et al., 2017). Furthermore, the Gumbel plots are based on 30 annual maxima (per season) and

therefore should be considered with some care.

### 4.1.3 Resolution analysis of the Mississippi basin

In the previous Sect. 4.1.2, our results show that a bias exist between simulated and observed basin-averaged precipitation for the Mississippi, especially in MAM ($\sim$ 0.5-1 mm d$^{-1}$, Fig. 5b). Moreover, no substantial differences in precipitation are found between the low and high resolution simulations (Fig. 5b). This is in contrast with our results for the Rhine basin, where better

precipitation estimates were found with the high resolution GCM, because of better resolved large-scale circulation patterns (Van Haren et al., 2015). Here, we will shortly assess the resolution sensitivity of large-scale circulation over the Mississippi basin in MAM. Furthermore, the role of convection over the basin is discussed.

First, analysis of the large-scale circulation patterns, bringing moisture from the Pacific over the Rockies, show similar results for the simulations and ERA-Interim (geopotential height at 500 hPa not shown). The advection of moisture from the






**Figure 8.** Gumbel plots of seasonal (DJF, MAM, JJA and SON) maximum 10-day precipitation sums [mm] over the Mississippi (left panels) and maximum discharge [m$^3$ s$^{-1}$] at Vicksburg (right panels) and their related return times T expressed in standardized Gumbel variate $x = -\ln(-\ln(T))$. Observed discharges are shown in black, high resolution forcing (T799) in red, low resolution forcing (T159) in blue and forcing with ERA-Interim in green. The discharge results are output from the 0.5° GHM.





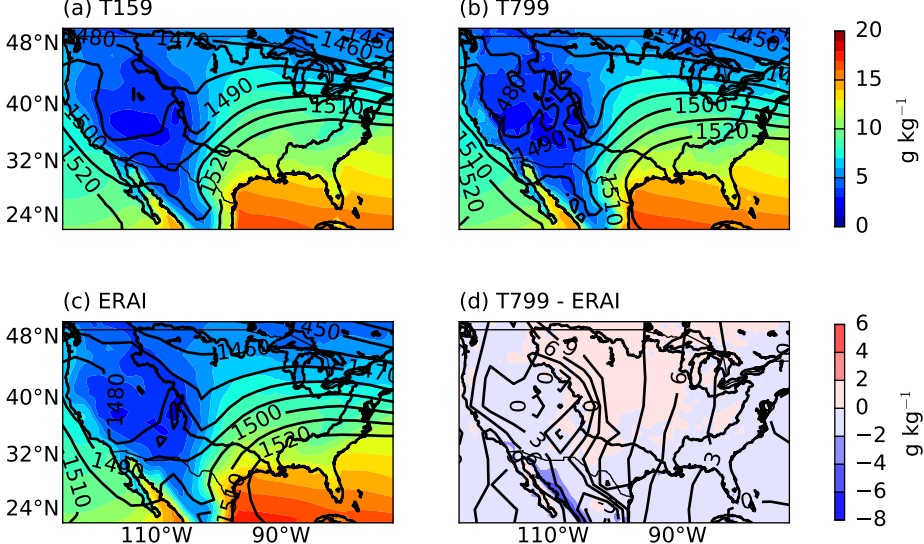

**Figure 9.** Averages of equivalent potential temperature [K] and geopotential height [$m^2$ $s^{-2}$] at 500 hPa for March, April and May over the Mississippi basin for a) the low resolution GCM (T159), b) high resolution GCM (T799), c) ERA-Interim (Observations) and d) the difference between high resolution (T799) and ERA-Interim.

Caribbean and Gulf of Mexico via the low level jet is captured by analysing geopotential height at 850 hPa and 2m specific humidity (Fig. 9). Only small differences in these variables are found between the simulations and ERAI (difference only shown for high-resolution, Fig. 9d). This indicates that the biases in precipitation in EC-Earth compared to observations and ERAI are not the result of differences in pre-conditioning of convective precipitation.

5    Convective cloud systems are smaller than the grid size of the model and therefore parameterized. We show the precipitation generated by the convective parameterization as monthly averages in Fig. 5b, to show the relative importance of convection in summer. In MAM, 55 % of the total precipitation budget is produced by this convection scheme. Fritsch et al. (1986) concludes that approximately 30 % to 70 % of the warm-season (April-September) precipitation between the Rocky Mountains and the Mississippi River can be related to mesoscale convective weather systems. The large contribution of convective precipitation

10   to total precipitation in the model likely explains why we do not find differences in basin-averaged precipitation between the two resolutions in MAM and summer (Fig. 5b). This is confirmed by Iorio et al. (2004) who found no improvements in precipitation over the USA in MAM and JJA with increased resolution, which was related to the dominance of convective precipitation in these two seasons. Balsamo et al. (2010) mentioned that large-scale weather systems in winter are easier to simulate in numerical weather predictions than convective systems in summer.



To summarize, this resolution analysis suggest that the large-scale circulation is well presented in both resolutions of the EC-Earth model. The large contribution of convection in the basin explains the similarity between the two resolutions. However, it does not yet explain the large bias with the observations.

## 4.2 Actual evaporation

### 4.2.1 Actual evaporation in the Rhine basin

Figure 5c shows that actual evaporation (Eact) from the simulations is overestimated compared to Eact from GLEAM, especially in winter (0.5 mm d$^{-1}$). This can be related to an overestimation of precipitation in winter, as increased precipitation can lead to larger evaporation rates. Actual evaporation from the high-resolution shows a smaller bias with observations than the low-resolution, which is consistent with our precipitation results. We also find an overestimation of Eact from ERAI compared to the reference GLEAM (Fig. 5c). However, precipitation in ERAI is not overestimated (Fig. 5a). The high evaporation amounts can explain the large underestimation of discharge from the GHM at Lobith, discussed in the next Sect. 4.3.1.

There is a large difference between actual evaporation directly from ERAI and actual evaporation indirectly from the GHM forced with ERAI. This difference is smaller for the EC-Earth simulations. Possibly, this is because of an improved land-surface scheme in EC-Earth (H-TESSEL), while ERAI is based on the old scheme (TESSEL) that does not contain a seasonal cycle in leaf area index and has a global uniform soil texture (Balsamo et al., 2009).

The yearly-averaged Eact of both simulations is comparable between the GHM and GCM, but there are seasonal differences (Fig. 5c). As both models (GCM & GHM) solve actual evaporation from the energy balance, these differences are related to the vegetation and soil characteristics of the models. Actual evaporation from the GHM is higher in the beginning of the year (January until June) and peaks earlier in the season compared to the GCM (Fig. 5). Overall, it seems that the Eact from the GCM is in better agreement with the reference GLEAM dataset.

### 4.2.2 Actual evaporation in the Mississippi basin

A consistent pattern between evaporation and precipitation is also found in the simulations for the Mississippi basin. The shift in seasonal cycle in the EC-Earth precipitation budget is reflected in a similar shift in the Eact budget (Fig. 5d). Furthermore, there are no substantial differences found in Eact between the two resolutions GCM. Nevertheless, we find large overestimations (∼ 0.5 mm d$^{-1}$) of Eact in winter (NDJF) in the simulations compared to the GLEAM dataset. In November and December, these overestimations can not be related to the precipitation budget. These high amounts of evaporation in winter where also found for the Rhine and are therefore possibly related to the performance of the GCM.

Actual evaporation from ERA-Interim directly is also largely overestimated compared to the GLEAM dataset, for the whole year. Betts et al. (2009) shows similar overestimations of actual evaporation in ERA-Interim over the Mississippi basin, compared to an observational dataset from Maurer et al. (2002). This can be related to the land-surface scheme (TESSEL) in ERA-Interim having a fixed leaf area index (van den Hurk et al., 2003) and a global uniform soil texture leading to low amounts of surface runoff (Balsamo et al., 2009), which could induce smaller amounts of interception and open water evapora-



tion resulting in overestimations of evaporation. Moreover, there are large differences in actual evaporation from ERA-Interim directly and from the GHM forced with ERA-Interim (Fig. 5d). These differences are larger for ERAI than for EC-Earth, which was also observed for the Rhine basin.

The actual evaporation from the GHM decreases faster from June onwards compared to the actual evaporation from the
GCM. A similar sudden decrease was found in the discharge at Vicksburg. In other words, there is a quick drying of the GHM from May/June. This should be mainly related to the vegetation and soil characteristics of the GHM as the GCM does not show the quick drying. Overall, for the Mississippi, it is hard to judge whether the evaporation product from the GCM or the GHM performs better in comparison with the observations as the seasonal bias in precipitation is also influencing the evaporation budget.

## 4.3 Discharge

### 4.3.1 Discharge in the Rhine

In Fig. 5e we show monthly-averaged discharge for Lobith with the two resolutions forcing data (GCM T799 & T159), ERA-Interim forcing, two resolutions GHM (0.5° and 0.05°) and the observations. Table 3 shows the four different discharge measures ($\overline{Q}_{\mathrm{mean}}, \overline{Q}_{\mathrm{max}}, \overline{Q}_{\mathrm{min}}, \overline{Q}_{\mathrm{peak}}$).

The discharge simulated with ERA-Interim forcing largely underestimates the observed discharge ($\sim 700$ m$^3$ s$^{-1}$), in particular from June until December (Fig. 5e). Photiadou et al. (2011) and Szczypta et al. (2012) present similar results, which they relate to an underestimation of precipitation in ERAI (Balsamo et al., 2010). However, our results show good estimates of basin-averaged precipitation from ERAI, except for a slight underestimation from August to November (Fig. 5a). Therefore we conclude that the GHM is too dry in the summer months of the Rhine basin, introducing a negative bias in discharge. We
also find lower discharges in the end of summer with EC-Earth forcing, possibly related to the dry bias of the GHM. From February until May, the overestimations in EC-Earth simulated precipitation are reflected in overestimations of discharge, with the largest bias for the low-resolution forcing (Fig. 5e & $\overline{Q}_{\mathrm{max}}$ in Table 3).

For the discharge extremes, we show similar Gumbel plots as for precipitation, now for annual maxima discharges per season (right panels in Fig. 6). The differences found in the return times of 10-day precipitation sums between the high and low
resolution simulations are reflected in the differences found in the return values for the discharge, in every season. However, the differences between simulations and observations are not clearly reflected from precipitation to discharge. Firstly, this is because the hydrological model has a large influence on the discharge results, which was already seen from the monthly average discharge plot. For example, the dry bias of the model results in lower discharge extremes in SON (Fig. 6h). Secondly, there is not a one-to-one correlation between precipitation sums and discharge. For example in spring (MAM) the discharge extremes
can be related to either rain, rain-over-snow or high temperatures that induce snow-melt. This could be the case for the highest return values of discharge in MAM which are underestimated by the model. In an assesment study of multiple GHMs and LSMs, it was also shown that W3RA performs less in snow-dominated regions (Beck et al., 2016a).





**Table 3.** Discharge measures ($\overline{Q}_{mean}, \overline{Q}_{max}, \overline{Q}_{min}, \overline{Q}_{peak}$) and the corresponding values for the observations and different model runs (0.5° and 0.05° W3RA GHM runs) with different forcing data (ERAI, EC-Earth T799 and EC-Earth T159) at Lobith.

| Measure | Obs. (1985-2014) | ERAI + 0.5° | T799 + 0.5° | T799 + 0.05° | T159 + 0.5° | T159 + 0.05° |
|---|---|---|---|---|---|---|
| $\overline{Q}_{mean}$ | 2 238 | 1 526 | 2 185 | 2 155 | 2 432 | 2 419 |
| $\overline{Q}_{max}$ | 7 028 | 4 206 | 5 055 | 4 610 | 5 875 | 5 468 |
| $\overline{Q}_{min}$ | 1 072 | 469 | 868 | 866 | 725 | 733 |
| $\overline{Q}_{peak}$ | Jan | Jan | Jan | Jan | Mar | Apr |

Overall, we can conclude that with the high-resolution EC-Earth forcing the seasonal cycle and the discharge characteristics (extremes) are better represented compared to the low-resolution forcing, mainly because of improvements in precipitation. The difference in precipitation between the model resolutions is clearly reflected in discharge, although biases in the hydrological model also influence these results.

We also tested the resolution sensitivity of the global hydrological model. We find small, but not significant differences in the discharge (measures) between the 0.5° and 0.05° model; the high resolution GHM (0.05°) gives slightly lower annual mean discharge results. With the 0.05° model, the peak flows are less extreme and the low-flows are similar to the low-flows from the 0.5° model. Because of a higher resolution orography a more detailed river network (ldd) is present in the high resolution model. Due to the presence of extra tributaries the response of precipitation to the main river may be damped, leading to a

decrease of the peak flow.

### 4.3.2    Discharge in the Mississippi

The seasonal variations in discharge are quite similar for the Rhine and Mississippi, with highest discharges at the end of the winter and the lowest at the end of the summer. We show the monthly averaged discharge at Vicksburg in Fig. 5f and the different discharge measures in Table 4.

We find an underestimation of the ERAI forced discharge during the whole year compared to the observed discharge. We can only partly explain this with the underestimation of ERAI precipitation in JJA (Fig. 5b). Precipitation from the ERAI LAND product agrees very well with the observations, however, discharge is still underestimated (data not shown). Therefore, we conclude that the underestimation in discharge is related to an overestimation of actual evaporation, which was shown in Sect. 4.2.2.

Annual mean discharge is underestimated ($\sim 2\ 000$ m$^3$ s$^{-1}$) with the low-resolution forcing and well simulated with the high-resolution forcing (Table 4). The monthly-averaged discharge forced with EC-Earth is too high in spring, because of too high precipitation values. In January and February, precipitation (including snow) is also overestimated in EC-Earth leading to increased discharges in April and May when the temperature rises. From May onwards the discharge decreases more rapidly in the model than observed.During the rest of the year, there is a clear discharge response to the precipitation budget. It is



**Table 4.** Discharge measures ($\overline{Q}_{\text{mean}}, \overline{Q}_{\text{max}}, \overline{Q}_{\text{min}}, \overline{Q}_{\text{peak}}$) and the corresponding values for the observations and different model runs (0.5° and 0.05° W3RA GHM runs) with different forcing data (ERAI, EC-Earth T799 and EC-Earth T159) at Vicksburg.

| Measure | Obs. (1985-2014) | ERAI + 0.5° | T799 + 0.5° | T799 + 0.05° | T159 + 0.5° | T159 + 0.05° |
|---|---|---|---|---|---|---|
| $\overline{Q}_{\text{mean}}$ | 18202 | 16286 | 18170 | 17672 | 16301 | 15656 |
| $\overline{Q}_{\text{max}}$ | 39626 | 37052 | 49106 | 48420 | 46061 | 44697 |
| $\overline{Q}_{\text{min}}$ | 4999 | 5548 | 4408 | 2597 | 3032 | 2382 |
| $\overline{Q}_{\text{peak}}$ | Mar | May | Apr | May | Apr | Apr |

possible that in October-November the improvements in discharge for the high-resolution exist for the wrong reason, as the second precipitation peak in the high-resolution is not seen in the observations.

For the extremes in SON, we also find a clear difference between the high and low resolution forcing (Fig. 8 g & h). With high-resolution forcing larger extremes are found, although discharge is still underestimated for lower return values, which was also found for the monthly averages. In DJF, there is a clear difference between the two resolutions for the largest return values in precipitation and this is also reflected in the return values for discharge, which are larger with high resolution forcing. In MAM, precipitation (extremes) are largely overestimated in EC-Earth which is reflected in slight overestimations of discharge in the lower return values but large overestimations for the larger return values (Fig. 8). It should be noted that W3RA does not contain reservoirs. This can lead to a faster reponse of discharge on precipitation in the model runs compared to the observed discharge. In the summer months (JJA), the discharge extremes are quite well represented by the model. Nevertheless, the ERAI forced discharge underestimates the extremes in these months.

In general, for the monthly averages and lower return values, the dry bias of the GHM is clearly reflected in the results. For the extremes with larger return values, we find that the signal of the precipitation extremes is reflected in the discharge extremes and the model performance plays a less important role. There are no substantial differences in discharge between the 0.5° and 0.05° resolution, as was also found for the Rhine.

## 4.4 Outlook on the extremes

In previous sections, we showed that, compared to observations, the mean (monthly) statistics of precipitation, actual evaporation and discharge are improved with high-resolution modelling. This is very relevant for assessing present and future weather. However, individual high-impact weather events, hydrometeorological extremes, are not discussed in this study yet. Realistic simulations of individual events are important in forecasting, impact studies and when assessing the potential effect of antropogenic climate change. In particular, the emerging field of event attribution requires that events are plausibly simulated with numerical models (Stott et al., 2013; Hazeleger et al., 2015). Here, we show two examples of extreme events and their representation within the models used in this study (GCM T799 → GHM 0.5°). A full analysis of events is beyond the scope of this paper, but these examples indicate that high resolution climate models are promising tools for event attribution and assessing future weather extremes.





**Figure 10.** Scatterplot for the Rhine basin of daily discharge [m$^3$ s$^{-1}$] with previous 10-day precipitation sums [mm] for the low resolution forcing (T159, left panel) and high resolution forcing (T799, right panel). The discharge results shown here are obtained with the 0.5° GHM. The different seasons are indicated with the colours and regression line and correlation value. The annual maxima of both 10-day precipitation sums and discharge are indicated with respectively the black squares and stars.




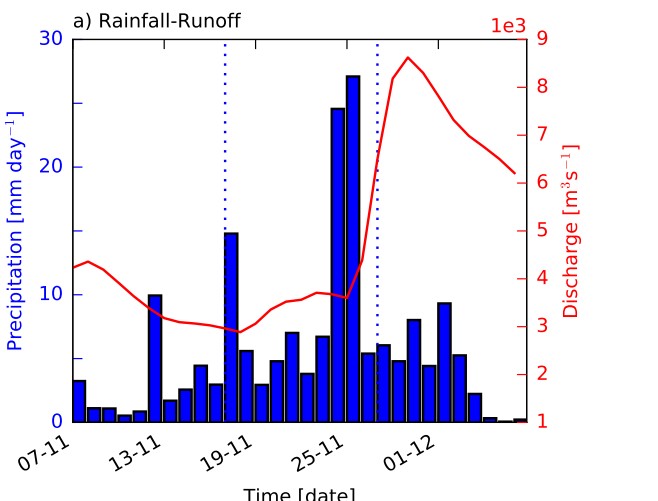
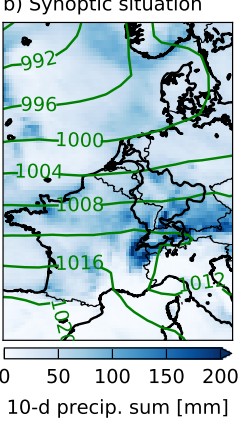

**Figure 11.** In the left panel precipitation and discharge for the Rhine are shown 20 days before and ten days after the selected event. The vertical dotted lines indicate the ten day period of which we took the sum of precipitation shown in the right panel. The contour lines indicate the mean sea level pressure in hPa averaged over these ten days.

First, we show scatterplots of daily discharge against previous 10-day precipitation sums. It should be noted that by applying a moving window over the 30-year timeseries, individual events are reflected in multiple subsequent data points. The stars and rectangles indicate respectively the annual maxima of 10-day precipitation sums and of discharge. The different colours indicate the occurring season (DJF, MAM, JJA and SON) for which we also show the linear fit and correlation value r.

5   The season with the highest correlation between precipitation and discharge is MAM, for both resolutions GCM over the Rhine basin. Initially we did not expect this, because snowmelt plays a role in this season as well, which would decrease the correlation. However, rain-on-snow events can lead to high discharges and often occur in MAM (McCabe et al., 2007). In addition, rain occuring over saturated soils can generate fast surface runoff during early spring as well.

The selected event for the Rhine basin is indicated with an open circle in Fig. 10b and is an annual maximum in discharge, 10   occurring at the end of November. The average discharge in SON is around $1\,500$ m$^3$ s$^{-1}$; In this case the discharge is almost $9\,000$ m$^3$ s$^{-1}$. Figure 11a shows the rainfall-runoff distribution from 20 days before until 10 days after the discharge event. In addition, the synoptic situation is shown with 10-day averaged mean sea level pressure and the 10-day precipitation sums (Fig. 11b). From the mean sea level pressure, we can infer that there is a low pressure system (mid-latitude cyclone) situated over the North Atlantic, before the coast of Norway, bringing moisture from the Atlantic over Europe leading to extreme 15   precipitation over the Alps. This case shows that a high-resolution GCM with a correct representation of the storm-track is needed to simulate such an extreme event.

For the Mississippi basin, we also find the highest correlation between 10-day precipitation sums and discharge during the spring season (MAM) for both resolutions forcing. Berghuijs et al. (2016) shows that during late winter and early spring



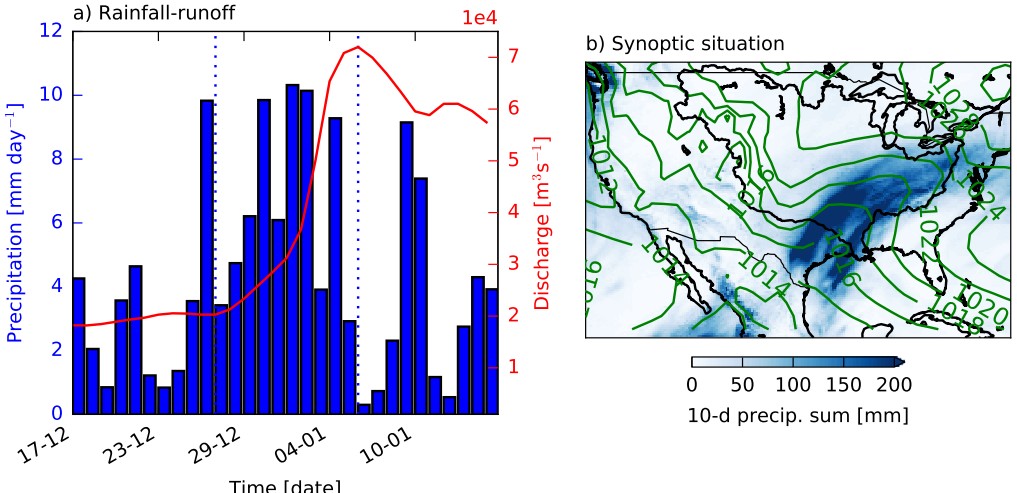

**Figure 12.** In the left panel precipitation and discharge for the Mississippi are shown 20 days before and ten days after the selected event. The vertical dotted lines indicate the ten day period of which we took the sum of precipitation shown in the right panel. The contour lines indicate the mean sea level pressure in hPa averaged over these ten days.

precipitation excess, defined as rainfall excess compared to available soil moisture storage capacity, is the most dominant flood generating mechanism in the USA. This process can explain the higher correlation in spring compared to the other seasons, as was also found for the Rhine basin. In SON, the correlation between 10-day precipitation sums and discharge is lowest, because during this season part of the precipitation will fall as snow which does not directly affects discharge.

The selected event over the Mississippi basin (open circle Fig. 10d) occurs in January and corresponds to both an annual maxima in the 10-day precipitation sum (66.8 mm) as well to an annual maxima in discharge (72 000 m$^3$ s$^{-1}$). In addition, the selected event is the second most extreme event in the DJF Gumbel plot for precipitation (Fig. 8a). From the synoptic situation (Fig. 12b) it seems that moisture is mainly transported from the Pacific and the Gulf of Mexico leading to precipitation over the South-East of the Mississippi basin, which is a region prone for extreme precipitation (Wehner et al., 2010). Berghuijs
et al. (2016) shows that (multiple) large precipitation events mainly occur over the South-East of the USA during winter. As the precipitation falls very close to Vicksburg, the response in discharge is relatively quick and leads to an exceptional high discharge (72 000 m$^3$ s$^{-1}$), with a return period of 30 years.

## 5   Discussion on methodology

In this study, we perform a one-way offline coupling of a GCM and a GHM (as illustrated in Fig. 3) to simulate the hydrological
cycle. To solve the land-surface hydrology, we have chosen to use a GHM rather then the GCMs land-surface model, as most



GCMs (including EC-Earth) do no include a routing module. Hence, routed discharge is not directly available for comparison with discharge observations. In addition, actual evaporation is calculated on a higher resolution in a GHM compared to a LSM.

To quantify the importance of both high resolution in the atmosphere and in hydrology, we performed a systematic comparison of high and low resolution simulations (Table 1). Such computations are time consuming and computationally expensive,

especially when we couple the high resolution GCM with the high resolution GHM. Therefore, this study is limited to one particular climate and one particular hydrological model. In addition, there are only few experiments available which run at such high resolution as EC-Earth T799 (Davini et al. 2017; Delworth et al. 2012; and more references within Haarsma et al. 2016). An advantage of the EC-Earth experiments is that we can compare different resolutions within the same set of physical parameterizations. As we are interested in the dependence of resolution, an in-depth analysis of the performance of W3RA is

beyond the scope of this study. Nevertheless, the results show that W3RA overestimates actual evaporation which results in an underestimation of discharge in both the Rhine and the Mississippi basin.

An increase in horizontal resolution results in an improved representation of the landscape, i.e. improved spatial heterogeneities in topography, soil and vegetation. This is beneficial for simulating hydrometeorological processes for both the GCM and the GHM. In case of an increased resolution GCM, large-scale dynamics will be resolved on smaller scales, while

mesoscale phenomena and moist and radiative processes are still parameterized. On the other hand, in GHMs, most relevant processess are parameterized. With an increased resolution GHM, processes which are not included at coarse resolution, such as lateral groundwater flow, need to be included at high resolution. However, to keep the consistency, we have taken into account the same set of parameterizations for both resolutions.There are also large uncertainties in the parameters itself, especially because most are non-physical and difficult to determine across scales (Melsen et al., 2016). Several studies apply parameter

regionalization techniques to adapt parameters across scales (Samaniego et al., 2010), but that goes beyond the scope of this study. To allow for a fair comparison between the different resolutions, we simply remapped the parameters from the low to the high resolution model, except for vegetation and orography which are known at high resolution (see Sect. 3.2). With this technique, it is possible to see the improvements in precipitation, due to higher resolution GCM, reflected in the discharge results.

**6   Summary and conclusions**

We study the benefits of increased spatial resolution from both the climate and hydrological perspective by coupling a global climate- and hydrological model. Thereby we analyse three main components of the hydrological cycle; precipitation, actual evaporation and discharge. We focus on two contrasting river basins with enough verification data but different climatic drivers; the Rhine and Mississippi basin.

By increasing the resolution of the EC-Earth GCM from $\sim 120$ km$^2$ to $\sim 25$ km$^2$, precipitation over the Rhine basin improves significantly, caused by the better represented large-scale circulation patterns. This is achieved by the combination of high resolution and global modelling. Therefore, we emphasize the need to invest in high resolution global simulations when studying climatic impacts on regions situated along the storm track.





The climatic drivers of the Mississippi basin are the large-scale systems from the Pacific, moisture transport from the Caribbean, possibly associated with tropical storms, and local convective events. No significant changes in the average precipitation budget over the Mississippi are found with increased resolution. Most likely because it is strongly dependent on the representation of small-scale convective processes, as the large-scale patterns are represented well in the simulations compared to reanalysis data. That is why, for the Mississippi basin, we recommend to use convection permitting models, to more explicitly resolve tropical storms and moist convective processes. For now, we conclude that the increased resolution GCM ($\sim 25$ km$^2$) is not as beneficial for simulating precipitation over the Mississippi basin as it is for the Rhine.

To increase the model resolution of the GHM, we have remapped the parameters from the 0.5° to the 0.05° resolution, except for orography and vegetation. With these settings for the high resolution GHM ($\sim 5$ km$^2$), no significant changes in discharge were found for both basins. Nevertheless, the (improved) precipitation from the GCM is reflected in (improved) actual evaporation and discharge from the GHM. That is, improved discharge and evaporation estimates are found for the Rhine basin when the GHM was forced with the high resolution GCM simulations. For the Mississippi basin, no substantial differences in discharge were found between the two resolutions input GCM and the two resolutions GHM for the Mississippi basin. Improvements in discharge are expected with high resolution GHMs when hydrological processes and parameters are better understood and described. For now we conclude that for a setup as ours, investing computational resources in an increased resolution GCM is more beneficial than in an increased resolution GHM when studying the impact of climate on hydrological processes, especially for basins situated along the mid-latitude storm-track.

## 7  Code availability

The code of the global hydrological model W3RA and the routing module are available via https://github.com/openstreams/wflow (wflow_w3ra and wflow_routing).

## 8  Data availability

Both the E-OBS and CPC precipitation datasets are freely available (from respectively http://www.ecad.eu/download/ensembles/ensembles.php and ftp://ftp.cdc.noaa.gov/Datasets/cpc_us_precip/). The ERA-Interim, ERA-Interim LAND and ERA20C reanalysis data are freely available from the ECMWF. The GLEAM dataset can be downloaded via https://www.gleam.eu/. The discharge data is obtained from the Global Runoff Data Centre and is freely available. The EC-Earth model output and the W3RA model output is available upon request by the author.





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
