# Peer review of "An evaluation of the importance of spatial resolution in a global climate and hydrological model based on the Rhine and Mississippi basin"

_Hydrology and Earth System Sciences, 2017_

## Referee Comment (RC1) · Anonymous Referee #1 · 3 Nov 2017

The study 'An evaluation of the importance of spatial resolution in a global climate and hydrological model based on the Rhine and Mississippi basin' by Benedict et al evaluates the effect of increasing resolution in a global climate model and a global hydrological model on the mean, seasonal cycle and extreme river discharge in the Rhine and Mississippi basins. I find this study very original. I particularly like the idea of cross-resolutions (high GCM - high GHM, high GCM - low GHM, low GCM - high GHM, low GCM - low GHM), and I think that the findings are well suited for the HESS journal.

The figures are clear. The paper is well written, but it could gain in clarity. Some

sections are not well organised, some important information (about the models for example) is missing. I also think that a few additional analyses could be added to better see the benefits of running high-resolution GCMs, compared to the low-resolution and particularly compared to observations.

Moreover, it is only in the discussion section that we learn that horizontal transport is switched off in the GHM. This is probably why there is no improvement between the low and high-resolution GHM, which probably biases the findings of this study. It would have been essential to see the impact of such improved representation of river flow on the mean and extreme discharge. See my detailed comments below.

Detailed comments:

- Page 2, L10: could you expand this sentence to specify which parameters are unknown or not easily quantifiable? Also in Page 5 L3 and L4: which parameters are remapped? Which method is used to remap them?

- Section 3.2: it is not mentioned whether there is horizontal transport in W3RA or if it is only vertical through the soil layers. According to the discussion section, the high-resolution version could have horizontal transport but it was switched off for better comparison with the low-resolution model. It seems to me that this should have been kept on for a more realistic representation of the moisture flow. Why can't the low-resolution GHM have it on as well? If horizontal transport was on in both versions, we would certainly see an impact of resolution, due to the slopes of orography for instance. Here the authors conclude that resolution does not play a role in the GHM, while (to me) a very important aspect of the model is left out.

If feasible, it would improve the study to have an extra simulation using the low-resolution GCM and low- and/or high-resolution GHM with horizontal transport. If not possible, then this aspect needs to be highlighted much more in the abstract, model description, and discussion.

- Page 5, L14-15: is one year enough for spinup? Is soil moisture in equilibrium? How deep are the soil layers?

- Section 3.4: The authors use the term 'coupling' between the GCM and GHM but the right term should be forcing/driving, as their is no interactions/feedbacks involved.

At which frequency are the forcing fields used?

- Page 6, L3: GHM forced with ERAI data: how long is the simulation? Is it 1 simulation of 30 years, or 6 simulations of 5 years?

- Page 8, L14-15: Is it really an improvement due to the storm track? Could it be also that the high-resolution GCM simulates precipitation over orography more accurately, as well as the dry shadow at the lee of the mountains (as shown on Fig 4)? To verify the hypothesis of better moisture transport, the authors could do a map of precipitation and moisture fluxes (as arrows) using a larger domain that includes part of North Atlantic.

It would be good also to add the convective part on this panel to determine if the peak in June is mostly convective.

- Page 9, Fig 5: evaporation panels: are solid lines GCM only, and dashed lines GHM at 0.5d forced by low and high-resolution GCM? If so, this needs to be made clearer in the caption.

- Page 11, section 4.1.2: why is the high-resolution GCM worse than the low-resolution for the most extreme precipitation events in SON, while the discharge is better?

Moreover, the high-resolution model shows much higher precipitation extremes but Fig 5 shows a similar mean seasonal cycle between low- and high-resolution models. So what is the contribution of extreme precipitation to the mean over the Mississippi?

- Fig 7: it is hard to see any difference. Do instead a difference plot. CPC, low-res minus CPC, high-res minus low-res. It could even be more informative to split it into seasonal means.

- Section 4: I find it difficult to have to jump back and forth between the different figures, read about the Rhine, then the Mississippi, then the Rhine again, etc. To facilite the understanding of the results and connect the processes together, it would be easier to have a whole section on the Rhine, then a whole one on the Mississippi, and modify the Figures accordingly.

- Fig 10: how do the models compare with observations? Is the distribution at high-resolution closer to observations?

Moreover, would these distribution look the same in the high-resolution GHM? If horizontal transport was allowed, how would these distributions look like?

- Page 20, L15-16: high-resolution is needed for such an extreme event, but is it realistic compared to observations? Adding observations would be useful here.

Moreover, if you select an extreme event in the low-resolution simulation, what does the large-scale circulation look like? The low-resolution model is probably able to simulate the large-scale pattern right but does not precipitate over the right location because of orography, or does not transport enough moisture across the ocean. It would be useful to have an extra case using the low-resolution model, look at the circulation, moisture transport and precipitation.

Minor comments:

- Page 2, L20: Replace Table 1 by Fig 1

- Page 2, L28-31: Add references for the Mississippi basin, as done for the Rhine basin before.

- Page 3, L2: replace 'empties' by 'discharges'

- Page 4, L13: replace 'medium' by 'low' for consistency

- Page 5, L7-8: instead of writing 'which means that there are no extra processes resolved at the higher resolution', which is vague (what extra processes? Horizontal

transport? Others?), add that differences between low and high-resolution GHM only come from better representation of orography and vegetation.

- Page 5, L9: replace Table by Fig

- Page 5, L12: what is ldd? (also page 17, L8)

- Section 3.3: replace title by a more explicit one, such as 'Observational datasets for model validation'

- Page 5, L29: replace ERA-Interim LAND by ERA-Interim/Land.

- Page 6, L2: replace Table by Fig

- Page 15, L26: replace where by are.

- Page 17, L5: replace but by and.

- Page 17, L18: replace weather by climate.

- Page 17, L23: replace arrow by words, for example: 'in the T799 GCM forcing the 0.5d GHM'.

- Fig 10: it looks like the selected events are annual means, as it is difficult to see the triangles. To see the events better, use larger triangles for example.

- Fig 10 caption: 'The different seasons are indicated with the colours and regression line and correlation value': be more descriptive: DJF (purple), etc. Be more descriptive on the plot as well: AM 10TP, AM Q?

- Page 20, L1: replace 'First, we show' by 'Fig 10 shows'

- Page 20, L3-4: repetition of caption, delete.

- Section 4.4: use subsections for the Rhine and the Mississippi.

- Page 21, L14: remove the term coupling

- Page 22, L6: replace 'experiments' by 'model simulations'

- Page 22, L32-33: cite previous studies, as this finding is not new.

- Page 23, L26: replace 'by the author' by 'to the authors'

- In many places in the manuscript, the word 'run' is used. I think 'simulation' is better in a scientific context, while 'run' pertains to the technical aspect of the simulation.

- Some acronyms are not defined: CMIP5, ISI-MIP, FAO

- I found a large number of typing errors in the manuscript. I suggest the authors to carefully check their manuscript. For example, there are many words in singular while they should be plural.

A few examples:

- title: use plural: models

- Page 2, L22: use plural: models

- Page 2, L24: use plural: basins (also in other places in the text)

- Page 2, L29: typo: Caribbean

- Page 2, L30: typo: precipitation

- Fig 2 caption: use plural: basins, stations

- Page 5, L5: typo: conclude

- Page 5, L14: typo: timeseries

- Page 11, L2: typo: particularly

- Page 11, L3: typo: overestimation

- Page 17, L15: use plural: resolutions

There are many others.

---

## Referee Comment (RC2) · Anonymous Referee #2 · 22 Nov 2017

This study compares the benefits of increasing the spatial resolution of a global climate model (from 120 to 25km) and of a global hydrological model (from 50 to 5km). The comparison is performed in Rhine and Mississippi basin. The benefits are variable and illustrate that an increase in resolution does not automatically lead to an increase in simulation realism. The study is interesting and the simulations performed are quite unique. However, it is not always clear how the models were adapted to be run at higher resolution, and the experimental setup has some flaws, which prevent the isolation of the benefits of the resolution increase. My major comments are on the spatial scale of the simulations (A), the benefits of increasing the resolution of the climate model (B) and of the hydrological model (C).

**Major comments**

A. Spatial scale: The authors take a GCM all the way to the regional scale, which is uncommon in hydrological studies and hence makes their study particularly interesting. They essentially skip the RCM step by running their GCM at a 25km resolution. I recognize this has advantages, in particular since RCMs typically cannot correct for errors in large-scale fields (P2 L16-17).

There are however two essential steps that require further discussion:

A1. It is debatable whether the resolutions the authors run the GCM at are high enough to capture the hydrological processes they are interested in. For instance, it is unclear whether the influence of elevation on snow can be correctly captured in mountainous areas. Snow processes are essential in the studied catchments, as the authors recognize: "[s]now melt, in combination with frozen soils, can occasionally lead to extreme flood events as well (Hegnauer et al., 2014)", and similarly in the Mississipppi basin, "[m]ost flood events occur in winter and spring due to heavy precipitation, snowmelt and rain-on-snow events." I understand that to bridge the resolution gap between the climate model and the hydrological model, the climate variables are "remapped" (P5 L9-10). I ask the authors to explain in more details how this is done, and in particular to discuss if this step provides climate simulations at a resolution high enough to enable them to capture the hydrological processes relevant for their study.

A2. Bias-correction is now routinely applied to climate model simulations before they are used for hydrological modelling. This allows for the adaption of the simulations to local conditions, and is usually necessary to produce realistic hydrological simulations. The authors might want to explicitly acknowledge that they skip this step.

Now, let us assess whether higher GCM resolution leads to better simulations:

B1. In Figure 4b) precipitation is overestimated in specific locations, in particular on the Italian side of the Alps. I am not convinced by the authors' explanation that this is
caused by the "underestimation of precipitation in the E-OBS dataset, which is based on a sparse gauge network in mountainous areas" (P8 L9-10). Those high precipitation locations are in unexpected places and I cannot think of atmospheric processes that could cause them and explain their highly localized pattern. It is my impression that those red/orange grid cells reflect errors in the model simulations at high resolution, and illustrate that increasing model resolution does not immediately lead to improved simulations.

B2. Increasing the resolution from T159 to T799 slightly improves the precipitation in the Rhine basin, but not in the Mississippi basin (Figures 5 a) and b)). The authors argue that this is partially explained by the imperfect "representation of small-scale convective processes" and "recommend to use convection permitting models, to more explicitly resolve tropical storms and moist convective processes" (P23 L5-7). I suggest that they provide evidence from other studies that further reducing the resolution will provide the better precipitation simulations in the Mississippi basin or similar areas. It is an expensive solution, hence it is important to better assess its chances of success based on literature already published.

B3. More details are needed to explain how the GCM was adapted to run at higher resolution. In particular, the authors state that "the land-use products are similar for the two resolutions, where the high-resolution information is used for the high-resolution runs", but it is unclear what the "products" are and in which respects they are "similar".

B4. Although I recognize the importance of getting large-scale processes right, is it necessary to run a climate model over the entire planet on a 25km grid to capture them adequately? As a follow-up to point A2, an alternative would be to run the GCM model at standard resolution (say 120km) and then use a convection permitting model over the area of interest. Because computational resources are scarce, I think this article would be more useful for the design of future experiments if those points were discussed, even though they go slightly beyond the scope of this paper.

Now, does higher GHM resolution leads to better simulations?

C1. "we remapped the parameters from the 0.5 to the 0.05 resolution" (P5 L5). It is essential to explain how the remapping was done, as it can significantly influence the perceived benefits of the increased resolution. Please dedicate a paragraph to it, explain how your design differ from Melsen et al. (2016) and discuss whether their results are truly transferable to your study. This is similar to my comments A1 and B3.

C2. Horizontal transport was switched off in the GHMs. I agree with reviewer 1 that this makes it difficult to truly assess the gains of the increased resolution. This aspect of the experimental setup should be made much more explicit and the implications for the results should be better discussed.

C3. I find Tables 3 and 4 useful to disentangle the effects of resolution increase in the GCM vs. in the GHM, but the results would be easier to interpret if they were represented graphically, for instance using barplots.

C4. I wonder how close GHMs are to replace calibrated catchment-scale hydrological models. To be able compare their performance, it would be useful to compute the NSE for hydrological simulations driven by ERA-interim. I recognize that it would not be a completely fair comparison, since the calibration of the catchment-scale model usually relies on observed discharge data, but it would be interesting to assess how well an uncalibrated GHM does in comparison.

**Minor comments**

"Increasing the resolution of a GHM also requires to increase the number of unknown, and often not easily quantifiable, model parameters" (P2 L9-10). This is also true for climate models, which also involve parameterizations whose parameters cannot be determined using measurements or physics, and hence have to be tuned.

"The representation of these processes is resolution dependent" (P2 L30-31). The representation of all processes is resolution dependent, I suggest removing this sentence.

"For both resolutions, six members of five years (2002-2006) are created, resulting in 30 years of data representing present climate" (P4 L14-15) Please explain why six simulations of five years were run instead of a single simulation of 30 years. Please also explain how the six members differ.

"For each member, we perform a spin-up of five years (length of timeserie per member). We leave out the first year for analysis due to the influence of spin-up, which results in 24 years of discharge simulations to analyse." (P5 L13-15) These two sentences apparently contradict each other: was the spin-up (i.e. the period not used in the validation phase) of 1 or 5 years?

I agree with the first reviewer that going back and forth between the Mississippi and the Rhine makes the text difficult to follow, and that it would be better to entirely describe the simulations in one catchment and then move on to the other one.

Figures 5, 6, 8, 10: Those Figures would be easier to understand if the authors took the legends out of the panels, synthesised them in a single legend and added it at the bottom of the Figure.

---

## Author Comment (AC1) · 14 Dec 2017

We would like to thank review referee #1 for the constructive comments on the manuscript. In this document we will clarify and respond to the comments. If we agree with a comment, we will not explicitly answer here, but adapt the text accordingly in the revised manuscript.

A major comment is on the horizontal transport in the GHM. There is no lateral redistribution of water between grid cells in both resolutions of the GHM W3RA. It is also not common to have a groundwater flow component in a global-scale model, although it has been implemented in some (de Graaf et al., 2015). Nevertheless, we will include in

the experimental set-up that lateral groundwater flow is not implemented, whereby we refer to the documentation of W3RA for a more detailed description of this assumption (Van Dijk, 2010). In the discussion, we mention that lateral groundwater becomes more and more important at higher resolutions, starting from 1 km (van Dijk, 2010; Bierkens et al., 2015; Wood et al., 2012). This sentence was meant as an outlook for future experiments but does not corresponds with our simulations, which we should clarify. In addition, there is horizontal transport of runoff via the routing module which is run after the hydrological model run. This routing scheme is run at different resolutions for the different resolutions GHM, namely: for the low resolution GHM (0.5 degree) we perform the routing on a resolution of 0.5 degree, for the high resolution GHM (0.05 degree) we perform the routing on a resolution of 0.0833 degree. This means that for the high resolution run we apply closest distance interpolation to remap runoff to the routing resolution.

Detailed comments:

* Reviewer: Page 2, L10: could you expand this sentence to specify which parameters are unknown or not easily quantifiable? Also in Page 5 L3 and L4: which parameters are remapped? Which method is used to remap them?

Response: On page2, line10 we could include an example of an unknown and not easily quantifiable variable but at this point in the introduction we think it is too specific to mention a whole list of parameters. In the experimental set-up we will include a reference to the model documentation (page 73-74) where one can find all the parameters which are included in W3RA and therefore will be remapped from the low to the high resolution (except for orography and vegetation). We remap the parameters from the low to the high resolution using the resample statement in PCRaster (Karssenberg et al., 2010).

* Reviewer: Page 5, L14-15: is one year enough for spin-up? Is soil moisture in equilibrium? How deep are the soil layers?
Response: For each member we performed a spin-up of five years (total length of member). Thereafter we perform the five year simulation, from which we leave out the first year for analysis. So in the end we have 6 years of spin-up, which we assume is long enough for soil moisture to be in equilibrium. Figure 1 shows a simulation of member 3 with the initial states of member 3 and member 4. From the figure we can see that in general, and especially after one year, the discharge signal is almost equal. This indicates that the discharge is hardly influenced by the initial states. The depth of the three soil layers are 0.15, 0.85 and 4 m (i.e. 0-0.15,0.15-1, 1-5 m).

\* Reviewer: Section 3.4: The authors use the term 'coupling' between the GCM and GHM but the right term should be forcing/driving, as there is no interactions/feedbacks involved.

Response: We agree with the referee that 'coupling' of the GCM and GHM is not the appropriate term in this study. Therefore we will replace coupling with forcing or driving in the revised manuscript.

\* Reviewer: Page 6, L3: GHM forced with ERAI data: how long is the simulation? Is it 1 simulation of 30 years, or 6 simulations of 5 years?

Response: We will clarify in the experimental set-up that for verification we force W3RA with 30 years of ERA-Interim data from 1984 until 2014. For all the simulations with forcing from ERA-Interim we have performed a spin-up from 1979-1990.

\* Reviewer: Page8, L14-15: Is it really an improvement due to the storm track? Could it be also that the high-resolution GCM simulates precipitation over orography more accurately, as well as the dry shadow at the lee of the mountains (as shown on Fig 4)? It would be good also to add the convective part on this panel to determine if the peak in June is mostly convective.

Response: We agree with the reviewer that the improvements in precipitation over the Rhine basin in the high-resolution GCM are likely a combination of an improved storm-
track, general circulation and a more detailed topography. The improvements related to topography are mentioned in L6-7, and not repeated in L14-15. We will mention topography again as a likely reason for improvement in L14-15. Figure 2 shows the contribution of convective precipitation to the total precipitation over the Rhine basin, peaking in June. This will also be shown in the revised manuscript.

\* Reviewer: evaporation panels: are solid lines GCM only, and dashed lines GHM at 0.5d forced by low and high-resolution GCM? If so, this needs to be made clearer in the caption.

Response: This is correct, we will clarify the caption.

\* Reviewer: Page 11, section 4.1.2: why is the high-resolution GCM worse than the low-resolution for the most extreme precipitation events in SON, while the discharge is better? Moreover, the high-resolution model shows much higher precipitation extremes but Fig 5 shows a similar mean seasonal cycle between low- and high-resolution models. So what is the contribution of extreme precipitation to the mean over the Mississippi?

Response: For the precipitation extremes in the Mississippi in SON, we do not see that the high-resolution GCM is worse than the low-resolution GCM in the most extreme precipitation cases. We think it is the other way around. Which than corresponds with higher extreme discharges with forcing from the high-resolution GCM. In SON we see much higher precipitation extremes in the high-resolution GCM (Fig. 8g). In these SON months we also find a higher monthly mean of precipitation over the catchment (Fig. 5b). We indeed see that the mean in DJF T799 and T159 are similar, although the extreme precipitation values are higher for the T799. We would like to mention that for the extreme precipitation plots we use 10-day precipitation sums. We can conclude here that the extremes in DJF do not influence the precipitation mean in this season.

\* Reviewer: Fig. 7: it is hard to see any difference. Do instead a difference plot. CPC, low-res minus CPC, high-res minus low-res. It could even be more informative to split

it into seasonal means.

Response: We agree that it is hard to see the differences between the high-res, low-res and CPC data, therefore we will adapt the colour scheme as can be seen in Fig. 2. Now the differences between the simulations appear more clearly. We do not change the colour scheme for the Rhine, as the current colour scheme shows clearly the differences for the Rhine. We agree that seasonal means will provide more information, nevertheless this also will increase the amount of figures. Therefore we will include the seasonal means in this review (Figure 3) and in the appendix.

* Reviewer: Difficult to jump back and forth between Rhine and Mississippi.

Response: We understand your comment on jumping back and forth between the Mississippi and Rhine. At first hand, we decided this set-up as want to compare the two different basins. Nevertheless, as the anonymous referee #2 also suggests to first discuss the Rhine and then the Mississippi we will do so if we get the opportunity to revise the manuscript.

*Reviewer: Fig 10: how do the models compare with observations? Is the distribution at high resolution closer to observations?

Response: We added the correlations between observed 10-day precipitation sums and observed discharge. We are still working on making this plot more clear.

*Reviewer: Page 20, L15-16: high-resolution is needed for such an extreme event, but is it realistic compared to observations? Adding observations would be useful here.

Response: The simulations of EC-Earth are not constrained by observations (except for the configuration of the model and the boundary conditions: sea surface temperature, greenhouse gases, aerosols and land use). Due to the chaotic nature of atmospheric flow it is not possible to make a one-to-one comparison of specific events between the simulations and observations (i.e. we can only compare the simulations and observations in a statistical sense).

**References**

Karssenberg, D., Schmitz, O., Salamon, P., de Jong, K., and Bierkens, M. F. P.: A software framework for construction of process-based stochastic spatio-temporal models and data assimilation, Environmental Modelling & Software 25(4), 489-502, 2010. doi: 10.1016/j.envsoft.2009.10.004

———————————————————————

**Fig. 1.** Discharge at Lobith and Vicksburg from member3 of the EC-Earth T159 data initialized with spin-up states from member4 (blue) and member3 (red line). The right plots are a zoom in of the left figures.

[Figure]

**Fig. 2.** Monthly averages of daily total precipitation (straight lines) and convective precipitation (dotted lines) over the Rhine basin for ERAI (green), EOBS (black), EC-Earth T799 (red) and T159 (blue).

[Figure]

**Fig. 3.** 30-year average of daily precipitation sums (mm d-1) over the Mississippi basin for the low resolution EC-Earth simulations (T159), the high resolution EC-Earth simulations (T799) and the CPC dataset

[Figure]

**Fig. 4.** Seasonal means (DJF, MAM, JJA and SON) of daily precipitation in mm d-1 from the low resolution GCM (T799, left) the high resolution GCM (T159, middle) and the observations (CPC, right)

[Figure]

**Fig. 5.** The original scatterplots of simulated 10-day precipitation sums and simulated discharge, where we now added the left plots with the correlation between the observational data.

---

## Author Comment (AC2) · 14 Dec 2017

We thank anonymous referee #2 for reading and commenting on our manuscript. We will reply on the major and minor comments which are raised.

(A) Spatial scale of the simulations

A1. Reviewer: I ask the authors to explain in more details how the remapping between the GCM and GHM is done, and in particular to discuss if this step provides climate simulations at a resolution high enough to enable them to capture the hydrological processes relevant for their study.

[Figure]

Response: First, we use closest distance spatial interpolation to remap the GCM variables to the resolution of the GHM. Second, we agree with the reviewer that the resolution of the high-resolution GCM cannot capture all relevant local hydrological processes, which is indeed a limitation of this study. We would like to emphasize that within this study we are interested in the impact of resolution of global models on hydrology. A resolution of 25 by 25 km is already at the high-end of state-of-the-art for global climate models and higher resolutions are hardly feasible, computationally, on a global scale.

A2. Reviewer: Explicitly acknowledge that there is no bias correction applied on the GCM simulations before they are used for hydrological modelling.

Response: For clarification, we will add to the experimental set-up that no bias correction is performed.

(B) Increasing resolution of the GCM

B1. Reviewer: Not convinced about the author's explanation that E-OBS shows underestimations of precipitation in the Italian Alps. It is my impression that those red/orange grid cells reflect errors in the model simulations at high resolution, and illustrate that increasing model resolution does not immediately lead to improved simulations.

Response: We do think that there is an underestimation of precipitation over the Alps in the E-OBS dataset. This is confirmed by Osnabrugge et al., 2017 (figure 4) who found large differences between EOBS and HYRAS (precipitation dataset from German Meteorological Service) in the Alpine area. More specifically, they found higher values for HYRAS compared to EOBS at the locations (Italian Alps) where we find an overestimation of EC-Earth compared to EOBS. Other studies also indicate that EOBS underestimates precipitation in the Alpine region (Turco et al., 2013; ). Finally, it should be noted that no under catch correction is applied in EOBS (Prein and Gobiet, 2016). We will include the above mentioned references in the manuscript to verify the underestimation of precipitation in the Alps in the EOBS dataset. Besides, topography is

extremely important for orographic precipitation. And we do agree that with the T799 resolution not all Alpine structure will be well captured, which we will include in the result section. Again, we would like to emphasize that the goal of this paper is to study the impact of horizontal resolution of global models, where the impact of changes in large-scale atmospheric drivers of precipitation is the most important effect.

B2. Reviewer: I suggest that they provide evidence from other studies that further reducing the resolution will provide the better precipitation simulations in the Mississippi basin or similar areas.

Response: The study from Liu et al. (2016) provides convection-permitting simulations with the Weather and Research Forecast model (WRF) over the USA for current and future climate. This model shows overall good performance capturing the seasonal precipitation climatology, except for a summer dry bias. In particular, snow is well simulated compared to snow observations (SNOTEL) (Liu et al., 2016). In these same simulations hourly precipitation from Mesoscale Convective Systems (MCSs) was detected and compared with radar-based precipitation estimates (Prein et al., 2017). They conclude that the convection-permitting simulations are able to capture the main characteristics and in particular the propagation of the MCSs (Prein et al., 2017). Above mentioned references will be included in the manuscript.

B3. Reviewer: More details are needed to explain how the GCM was adapted to run at higher resolution, especially for the land-use products.

Response: EC-Earth is based on IFS cy31r1. An extensive description of the model and it's land-surface characteristics can be find in the cited paper describing the simulations (Haarsma et al., 2013), describing the model (Hazeleger et al., 2010, 2012) and in the documentation of IFS (ECMWF, IFS Documentation Cy31r1, 2006). The input climate fields (i.e. land-surface characteristics) can be found in the documentation, they are similar for both resolutions and interpolated to the requested target resolution, in this case either T159 or T799. The land-use products are based on the Global Land

Cover Characteristics (GLCC) data (which are derived from Loveland et al., 2000). We will include the reference to the IFS documentation (ECMWF, IFS Documentation Cy31r1, 2006).

B4. Reviewer: Although I recognize the importance of getting large-scale processes right, is it necessary to run a climate model over the entire planet on a 25km grid to capture them adequately?

Response: We would like to emphasize the importance of running a GCM at high resolution (T799 instead of T159), for better representing large-scale circulation. Previous studies show that the large-scale circulation patterns significantly improve when the resolution increases from T159 to T511 (Jung et al., 2011) and from T159 to T799 (van Haren et al., 2015). The study of Jung et al. (2011) also shows that increasing the resolution of a GCM from T1279($\sim$16 km) to T2047($\sim$10km) leads to relative small changes. Moreover, teleconnections, in particular from the tropics, are important for weather regimes in mid-latitudes. A biased background state will affect extremes (e.g. Henderson et al., 2017). That is why we argue that running a convection-permitting model with the T159 GCM as boundary conditions is not the perfect design for future experiments. The literature mentioned above shows evidence that running a convection-permitting model with boundary conditions from the high-resolution GCM (T799) is a much better design for future experiments than downscaling from current global climate models, which we will include as an outlook in the manuscript. However we also like to stress that this study is focussed on global models and that we can only give an outlook on the use of convection-permitting models.

(C) Increasing resolution of the GHM

C1. Reviewer: Explain how the remapping of the hydrological parameters to the high resolution was done and discuss whether the results of Melsen et al. (2016) are truly transferable to your study.

Response: We remap the parameters from the low to the high resolution using the resample statement in PCRaster (Karssenberg et al., 2010). We will include a reference to the W3RA documentation for the list of parameters which are resamples from low (∼50 km) to high (∼5 km) resolution. To validate this approach of resampling the parameters towards the high resolution we refer to Melsen et al. (2016), who concludes that parameters are to a large extent transferable between different spatial scales. They base this conclusion on a sensitivity study of parameter transferability over resolution (∼50, 10,5 & 1 km) and time for the Thur basin in Switzerland. Although the catchment in their study is much smaller than the basins in this study, the change of spatial resolution from 50 to 5 km is comparable.

C2. Reviewer: Because horizontal transport was switched off in the GHM this makes it difficult to assess the gains of increased resolution, which should be made more explicit and better discussed.

Response: There is no lateral redistribution of water between grid cells in both resolutions of W3RA. It is also not common to have a groundwater flow component in a global-scale model, although it has been implemented in some (de Graaf et al., 2015). We mention in the discussion that lateral groundwater becomes more and more important at higher resolutions, starting from 1 km (van Dijk, 2010; Bierkens et al., 2015; Wood et al., 2012). This sentence was meant as an outlook for future experiments but does not corresponds with our simulations, which we should clarify. In addition, there is horizontal transport of runoff via the routing module which is run after the hydrological model run. We will clarify this in our experimental set-up.

C3: Reviewer: Change table 3 and 4 to barplots.

Response: See Figure 1 and Figure 2.

C4: Reviewer: I wonder how close GHMs are to replace calibrated catchment-scale hydrological models, for example by computing the NSE.

Response: For specified catchment studies, we advise to use calibrated catchmentscale models. However, the purpose of this study was to test the sensitivity of discharge to the resolution of a global hydrological model. For interpretation we have chosen to analyse to large catchments. For the Rhine we can compare our model results of ERA-Interim and W3RA with model results of ERA-Interim and HBV (Photiadou et al., 2011). For the discharge at Lobith, Photiadou et al. (2011) obtained a Nash-Sutcliff efficiency (NSE) of -0.54. The NSE for our simulations are shown below. For the Mississippi we are not aware of a catchment-scale hydrological study with ERA-Interim forcing to compare our results.

NSE:

T799 + 0.5°: -0.51

T799 + 0.05°: -0.49

T159 + 0.5°: -0.78

T159 + 0.05°: -0.72

Minor comments

If we do not discuss the minor comment stated by the reviewer it means that we agree with the reviewer and we will adapt this in the manuscript.

* Reviewer: Please explain why six simulations of five years were run instead of a single simulation of 30 years. Please also explain how the six members differ.

Response: The GCM experiments were at first hand performed for multiple research questions, like the impact of climate change on teleconnection responses to specific tropical sea surface temperature patterns (Haarsma et al., 2013). Namely, similar experiments for present climate (2002-2006) are also performed for future climate (2094-2098). These research questions motivated the larger ensemble approach of shorter runs. We will add in the manuscript that the research questions discussed in this paper could also be studied with a fewer longer runs.

In this study we use 5-year 6-member ensemble simulations from 2002-2006. A 10-year spin-up run at the low resolution (T159) was made, followed by a 9 month run (from January to October) spin-up run at the desired resolution. The 6 member ensemble was made by taking the atmospheric state of one of the first 6 days of October as initial state for each member. After this spin-up the spread in the atmospheric state was sufficient to treat the 6 runs as independent members (PhD thesis Ronald van Haren, page 65). For this extensive explanation on the difference between the six members we refer to Haarsma et al. (2013).

\* Reviewer: Difficult to jump back and forth between Rhine and Mississippi.

Response: We understand the comment on jumping back and forth between the Mississippi and Rhine. At first hand, we decided this set-up as want to compare the two different basins. Nevertheless, as the anonymous referee #1 also suggests to first discuss the Rhine and then the Mississippi we will do so if we get the opportunity to revise the manuscript.

\* Reviewer: Was the spin-up (i.e. the period not used in the validation phase) of 1 or 5 years?

Response: The spin-up period was 6 years (5 +1 year). We will make this more clear in the revised manuscript. A validation of the spin-up is shown in the reply to referee number 1.  

References

De Graaf, I.E.M., Sutanudjaja, E.H., Van Beek, L.P.H. and Bierkens, M.F.P., 2015. A high-resolution global-scale groundwater model. Hydrology and Earth System Sciences, 19(2), pp.823-837.

Haarsma, R.J., Hazeleger, W., Severijns, C., Vries, H., Sterl, A., Bintanja, R., Oldenborgh, G.J. and Brink, H.W., 2013. More hurricanes to hit western Europe due to global warming. Geophysical Research Letters, 40(9), pp.1783-1788.

Van Haren, R., Haarsma, R.J., Van Oldenborgh, G.J. and Hazeleger, W., 2015. Resolution dependence of European precipitation in a state-of-the-art atmospheric general circulation model. Journal of Climate, 28(13), pp.5134-5149.

Henderson, S.A., Maloney, E.D. and Son, S.W., 2017. Madden-Julian Oscillation Pacific teleconnections: The impact of the basic state and MJO representation in General Circulation Models. Journal of Climate, (2017).

Jung, T., Miller, M.J., Palmer, T.N., Towers, P., Wedi, N., Achuthavarier, D., Adams, J.M., Altshuler, E.L., Cash, B.A., Kinter Iii, J.L. and Marx, L., 2012. High-resolution global climate simulations with the ECMWF model in Project Athena: Experimental design, model climate, and seasonal forecast skill. Journal of Climate, 25(9), pp.3155-3172.

Liu, C., Ikeda, K., Rasmussen, R., Barlage, M., Newman, A.J., Prein, A.F., Chen, F., Chen, L., Clark, M., Dai, A. and Dudhia, J., 2017. Continental-scale convection-permitting modeling of the current and future climate of North America. Climate Dynamics, 49(1-2), pp.71-95.

Loveland, T. R., Reed, B. C., Brown, J. F., Ohlen, D. O., Zhu, Z., Youing, L. and Merchant, J. W. (2000). Development of a global land cover characteristics database and IGB6 DISCover from the 1km AVHRR data. Int. J. Remote Sensing, 21, 1303–1330.

Van Osnabrugge, B., Weerts, A. H., & Uijlenhoet, R., 2017. genRE: A Method to Extend Gridded Precipitation Climatology Data Sets in Near Real‐Time for Hydrological Forecasting Purposes. Water Resources Research.

Prein, A.F. and Gobiet, A., 2017. Impacts of uncertainties in European gridded precipitation observations on regional climate analysis. International Journal of Climatology, 37(1), pp.305-327.

Prein, A.F., Liu, C., Ikeda, K., Bullock, R., Rasmussen, R.M., Holland, G.J. and Clark, M., 2017. Simulating North American mesoscale convective systems with a convection-permitting climate model. Climate Dynamics, pp.1-16.

Turco, M., Zollo, A. L., Ronchi, C., Luigi, C. D., & Mercogliano, P. (2013). Assessing gridded observations for daily precipitation extremes in the Alps with a focus on northwest Italy. Natural Hazards and Earth System Sciences, 13(6), 1457-1468.

[Figure]

[Figure]

**Fig. 1.** Discharge measures (Qmean, Qmin and Qmax) for the observations and different model runs (0.5° and 0.05° W3RA GHM runs) with different forcing data (ERAI, EC-Earth T799 and T159) at Lobith.

**Fig. 2.** Discharge measures (Qmean, Qmin and Qmax) for the observations and different model runs (0.5° and 0.05° W3RA GHM runs) with different forcing data (ERAI, EC-Earth T799 and T159) at Vicksburg.